# Regulatory Effects of ABA and GA on the Expression of Conglutin Genes and *LAFL* Network Genes in Yellow Lupine (*Lupinus luteus* L.) Seeds

**DOI:** 10.3390/ijms241512380

**Published:** 2023-08-03

**Authors:** Natalia Klajn, Katarzyna Kapczyńska, Paweł Pasikowski, Paulina Glazińska, Hubert Kugiel, Jacek Kęsy, Waldemar Wojciechowski

**Affiliations:** 1Department of Plant Physiology and Biotechnology, Faculty of Biological and Veterinary Sciences, Nicolaus Copernicus University, Lwowska 1, 87-100 Torun, Poland; paulina.glazinska@umk.pl (P.G.); kesy@umk.pl (J.K.); 2Department of Immunology of Infectious Diseases, Hirszfeld Institute of Immunology and Experimental Therapy, Polish Academy of Sciences, Weigla 12, 53-114 Wroclaw, Poland; katarzyna.kapczynska@hirszfeld.pl; 3Life Sciences and Biotechnology Center, Łukasiewicz Research Network–PORT Polish Center for Technology Development, Stabłowicka 147, 54-066 Wroclaw, Poland; p.pasikowski@captortherapeutics.com; 4Captor Therapeutics S.A., Duńska 11, 54-427 Wroclaw, Poland; 5LABcenter Life Agro Biotechnology Ltd., Gliniana 14, 97-300 Piotrków Trybunalski, Poland; hubert.kugiel@generiks.pl (H.K.); wwojc72@gmail.com (W.W.)

**Keywords:** δ-conglutin, β-conglutin, *LAFL*, ABA, GA, yellow lupine, seed development, protein accumulation, gene expression, nanoLC-MS/MS, RNA-seq

## Abstract

The maturation of seeds is a process of particular importance both for the plant itself by assuring the survival of the species and for the human population for nutritional and economic reasons. Controlling this process requires a strict coordination of many factors at different levels of the functioning of genetic and hormonal changes as well as cellular organization. One of the most important examples is the transcriptional activity of the *LAFL* gene regulatory network, which includes LEAFY *COTYLEDON1* (*LEC1*) and *LEC1*-*LIKE* (*L1L*) and *ABSCISIC ACID INSENSITIVE3* (*ABI3*), *FUSCA3* (*FUS3*), and *LEC2* (*LEAFY COTYLEDON2*), as well as hormonal homeostasis–of abscisic acid (ABA) and gibberellins (GA) in particular. From the nutritional point of view, the key to seed development is the ability of seeds to accumulate large amounts of proteins with different structures and properties. The world’s food deficit is mainly related to shortages of protein, and taking into consideration the environmental changes occurring on Earth, it is becoming necessary to search for a way to obtain large amounts of plant-derived protein while maintaining the diversity of its origin. Yellow lupin, whose storage proteins are conglutins, is one of the plant species native to Europe that accumulates large amounts of this nutrient in its seeds. In this article we have shown the key changes occurring in the developing seeds of the yellow-lupin cultivar Taper by means of modern molecular biology techniques, including RNA-seq, chromatographic techniques and quantitative PCR analysis. We identified regulatory genes fundamental to the seed-filling process, as well as genes encoding conglutins. We also investigated how exogenous application of ABA and GA_3_ affects the expression of *LlLEC2*, *LlABI3*, *LlFUS3,* and genes encoding β- and δ-conglutins and whether it results in the amount of accumulated seed storage proteins. The research shows that for each species, even related plants, very specific changes can be identified. Thus the analysis and possibility of using such an approach to improve and stabilize yields requires even more detailed and extended research.

## 1. Introduction

Various species of lupine with sweet seeds (such as yellow, narrow-leaved, or white lupine) have been utilized more and more frequently in the food industry as a source of vegetable protein. Lupine seeds, in addition to high protein content and low content of anti-nutritional ingredients, also have numerous nutraceutical properties, such as increasing satiety with reduced energy intake and lowering blood pressure and glucose levels, as well as cholesterol and triglyceride levels, and play a significant role in combating cardiovascular diseases [1,2,3,4]. The main fractions of storage proteins in lupine seeds are albumins and globulins. Once found, they were called conglutins and divided into four groups: α, β, γ and δ. These proteins are encoded by small gene families, respectively: *ALPHA1*-*3*, *BETA1-7*, *GAMMA1-2* and *DELTA1-4* [5,6,7]. In order for the expression of genes encoding storage proteins to be activated, a number of events regulated by the interaction of various genes must occur; most often the genes in question encode proteins that possess the features of transcription factors [8,9,10].

The process of storage protein synthesis, so far best described for *Arabidopsis thaliana*, takes place during the seed maturation phase. The main regulators of these changes are transcription factors of the LAFL network, namely: LEAFY COTYLEDON 1 (LEC1) and LEC1-Like (L1L), ABSCISIC ACID INSENSITIVE3 (ABI3), FUSCA3 (FUS3), and LEC2 [11,12,13,14]. ABI3, FUS3, and LEC2 belong to the family of transcription factors binding the B3 domain, which is responsible for binding regulatory elements in the DNA [15,16]. LEC1, on the other hand, is a member of the NF-YB protein family. It interacts with the NF-YC and NF-YA subunits to form the NF-Y transcription factor. This allows it to bind DNA sequences and form a cis-element binding complex (CCAAT) involved in the initiation of the transcription [8,11,17].

During the seed maturation, *LAFL* genes show temporal expression patterns [18]. The protein encoded by *LEC1* is a key factor controlling embryonic development [19,20]. During late embryogenesis, its expression is inhibited by PICKLE (PKL) activity. This peptide belongs to the family of CHD3 factors responsible for chromatin remodeling [21]. These peptides are also involved in the regulation of transcription in many developmental processes of *Arabidopsis*. However, in this case, PKL exhibits nucleosome remodeling activity leading to the trimethylation of histone H3 lysine 27 (H3K27me3) [22,23]. When the amount of PKL protein decreases, *LEC1* becomes transcriptionally active again, allowing the activation of the rest of the genes of the *LAFL* network [19,22]. LEC2 positively regulates LEC1 [24], while LEC1 and FUS3 positively regulate *L1L* expression [25]. In turn, the expression of *ABI3* and *FUS3* is autoregulated and shows mutual activation [26,27,28].

ABI3, FUS3 and LEC2 bind to the RY motifs in the promoters of genes encoding storage proteins by means of their B3 domain, leading to the initiation of their expression [22,26,29,30]. Termination of the seed-filling phase is associated with a decrease in the transcriptional activity of *LEC1* and *LEC2* genes and with partial contribution by the presence of the VAL1 protein [22,31,32]. The VAL1 protein exhibits the ability to bind the same RY regulatory elements as described above *LAFL* in exception to *ABI3* [31]. Competition for the binding site inhibits the expression of *LAFL* genes. At the same time, the amount of PKL increases, which again causes changes in the chromatin structure, resulting in the inhibition of LEC1 transcription. Then, the amount of the LAFL group of proteins decreases, resulting in the inability to further synthesize storage proteins [21,22,33].

Research conducted over the last few decades shows that, apart from specific regulatory proteins, such as LAFL and PKL or VAL1, the participation of phytohormones in the control of seed filling processes is also very significant [34,35]. The most important role is attributed to the interdependence between gibberellic acid (GA) and abscisic acid (ABA) [36,37,38]. However, as numerous studies have shown, ABA especially may interact with ethylene, jasmonic acid or brassinosteroids [37,38,39,40]. In many plant species, phytohormones have been shown to affect the expression of genes related to the regulation of storage-protein accumulation, especially *FUS3* and *ABI3* [41,42].

In *A. thaliana*, activation of the FUS3 protein results in a decrease in the transcriptional activity of the gibberellin biosynthesis genes *AtGA3ox1* and *AtGA20ox1*, leading to an increase in the amount of ABA in developing seed cells [43,44]. In turn, ABI3, together with LEC1, LEC2 and FUS3, mediates ABA biosynthesis in developing seed tissues [37]. Studies conducted on *abi3*, *lec1*, *lec2*, and *fus3* mutants showed that defects in any of these genes caused severe abnormalities in maturing seeds, also revealing some common phenotypes, such as reduced ability to enter dormancy and reduced expression of genes coding storage proteins [45,46,47].

Knowing the principles of the general model of the accumulation of storage proteins in seeds, which has been described mainly in the model plant *Arabidopsis thaliana*, it is possible to identify several key factors necessary for the process to occur. However, data on the molecular mechanism of these transformations in many economically viable crop species remain incomplete. Taking into account all the problems arising from climate change that indirectly affect a variety of populations and industries, especially the field of agriculture, it becomes extremely important to know and understand the molecular transformations underlying the process of accumulation of storage proteins in lupine seeds. Yellow lupine deserves special mention as it accumulates the highest amounts of storage proteins among the previously mentioned lupins [48,49,50]. Nevertheless, it is also important to describe the mechanisms of interactions between the various components of the seed-filling process and to check their efficiency and the possibility of regulating the process through phytohormone applications, for example. Therefore, the aim of our study was to check whether the exogenous application of ABA and GA can significantly modify the expression of *LAFL* genes and thus affect the amount of accumulated storage proteins in maturing yellow lupin seeds. Undertaking such research is important because of the prospective applications in practice and the possibility of modulating molecular processes through exogenous application of phytohormones, for example. Conducting an experiment in natural cultivation conditions introduces a considerable number of complications caused by the variability of the conditions, but it allows one to more accurately trace the entire range of changes that may occur under the influence of additional factors, e.g., ABA or GA application. The collected data also show how many additional factors not taken into account in controlled conditions can act as superior controllers of the course of a specific process or modulate the response to the introduced factor.

## 2. Results

### 2.1. Identification of Homologs of Genes Encoding Conglutins and LAFL Genes

The transcriptomes were identified by means of high-throughput sequencing (NGS), and cDNA libraries created on mRNA templates found in yellow lupine cells and the expression of *LlBETA* and *LlDELTA2* genes as well as *LlLEC2*, *LlABI3* and *LlFUS3* genes were determined at 10, 20 and 30 days after anthesis (DAA) (GSE207091). The conducted experiments confirmed the presence of almost identical genes previously described in other *Fabaceae* species [6,7,51,52].

### 2.2. Expression of Genes from the LlLAFL Network in the Study of Transcriptomes

The use of high-throughput RNA sequencing made it possible to determine changes in the level of expression of genes involved in the regulation of the seed-maturation phase and encoding β- and δ-conglutin in the seeds of yellow lupine of the Taper variety. Expression levels were determined based on RPKM (reads per kilobase per million mapped reads) parameter values.

#### 2.2.1. *LlLEC2* Expression Level

The expression pattern of the *LlLEC2* gene (Figure 1a) shows an increase in transcriptional activity between days ten and twenty after anthesis, reaching a value over 243 times more ($p=4.79\times 10^{-94})$. In the last examined stage of the seed development (30 DAA), an almost 80-fold decrease in *LlLEC2* gene expression was observed compared to the previous day of development ($p=2.84\times 10^{-36}$).

#### 2.2.2. *LlABI3* Expression Level

The RNA-seq analyses performed indicate an increase in the amount of *LlABI3* transcripts in the successive days of seed development (Figure 1b). The transcriptional activity of the *LlABI3* gene on 20 DAA increased almost 157-fold compared to the baseline value ($p=5.65\times 10^{-107}$). On the other hand, on day 30, the relative expression level of the *LlABI3* gene was approximately 0.78-times higher compared to the 20 DAA variant ($p=2.75\times 10^{-29}$).

#### 2.2.3. *LlFUS3* Expression Level

The transcriptional activity of the *LlFUS3* gene determined in the RNA-seq experiment (Figure 1c) indicates an increase in the amount (over 210-times) of mRNA of the tested gene on 20 DAA compared to the value of 10 DAA ($p=3.39\times 10^{-78}$). In contrast, ten days later (30 DAA), there was a decrease (almost five times without statistical importance) in the expression level of the *LlFUS3* gene compared to the value of the previous variant (p=0.31).

#### 2.2.4. *LlBETA* Expression Level

The identified expression pattern of genes encoding β-conglutins indicates an increase in transcriptional activity at the last stage of the seed development studied (30 DAA) (Figure 1d). On 20 DAA, the expression level of *LlBETA* genes reached a value almost three times higher (2.95) without statistical significance when compared to the 10 DAA variant ($p=1.54\times 10^{-1}$). On the other hand, on day 30, there was an almost 1850-fold increase in the number of reads of genes encoding β-conglutins ($1.24\times 10^{-9}$).

#### 2.2.5. *LlDELTA2* Expression Level

The gene encoding δ-conglutin also showed the highest relative level of expression on day 30 after anthesis (Figure 1e), similarly to the case of *LlBETA* genes (Figure 1d). On 20 DAA, the amount of mRNA of the examined gene was over two times higher (2.14) compared to the previous stage of seed development (p=0.008). However, after the next ten days had passed, it increased over 1000-times (1903) ($p=1.51\times 10^{-87}$).

### 2.3. Changes in the Level of Expression of LlLAFL Genes and Genes Encoding β- and δ-Conglutins under the Influence of ABA or GA_3_ Application

As a result of the conducted real-time PCR experiments, the expression pattern of the identified genes was determined in the following days of the development of yellow lupine seeds of the Taper variety (15, 20, 30 DAA) under control conditions and under the influence of ABA or GA_3_ application at the designated hours of material collection (0 h, 4 h, 8 h). To verify the statistical significance of the results of the real-time PCR experiments, an analysis of variance was performed in a mixed model (3 × 3 × 3). The within-group variable was the time of harvest (0 h vs. 4 h vs. 8 h), while the between-subject variables included administered phytohormone (GA_3_ vs. ABA vs. Control) and stage of seed development (day 15 vs. 20 vs. 30).

#### 2.3.1. Relative Expression Level of *LlLEC2*

The analysis of differences in the level of expression of the *LlLEC2* gene (Figure 2) showed a statistically significant main effect of the developmental stage: $F(2.12=1644.97;\;p<0.001;\;\eta^{2}=0.99$. This means that the reduction in the transcriptional activity of the *LlLEC2* gene was independent of the phytohormones used and the time when the material was collected. On the other hand, the stage of seed development showed that in the successive studied periods, there was a significant decrease in the expression of the *LlLEC2* gene. In each of the described cases, the level of significance was lower than p=0.05.

#### 2.3.2. Relative Expression Level of *LlABI3*

Analysis of differences in the level of transcriptional activity of the *LlABI3* gene (Figure 3) showed a statistically significant interaction effect between all factors, $F\left(8.24\right)=3.30;p=0.011;\;\eta^{2}=0.52$. The expression level of the *LlABI3* gene showed significant linear differences on each successive day of seed development. It was shown to be independent of the hour and the phytohormone applied.

In each group of the seeds and at each hour of harvesting, there was a higher level of transcriptional activity of the *LlABI3* gene, and these correlations were statistically significant, taking a value of at least p<0.05. Analyzing the differences between the tested seed variants on 15 DAA (Figure 3), there was only a statistically significant decrease in *LlABI3* gene expression eight hours after ABA administration compared to seeds from the control group (p=0.017). On 20 DAA at zero hour, there was a statistically significant increase in *LlABI3* gene expression between seeds of exogenous GA-treated and control plants (p=0.017) (Figure 3). In addition, a significantly higher level of transcriptional activity of the tested gene was observed among the seeds of GA-treated plants at the eighth hour of harvest compared to the seeds of ABA-treated (p=0.011) and control (p=0.029) plants. In seeds with 30 DAA at the beginning of the sampling (0 h), a significantly higher level of *LlABI3* gene expression was observed after GA_3_ application compared to the seeds of plants after ABA application (p=0.038) and controls (p=0.002). At the 4th hour of harvest, a statistically significant, higher level of *LlABI3* gene transcript was observed only in the seeds of plants after GA_3_ application compared to the seeds of plants treated with ABA (p=0.040). A similar effect was observed at the eighth hour of harvest, where a higher level of *LlABI3* gene expression was noticeable in the seeds of plants after GA_3_ application than after ABA application (p=0.001). Moreover, the seeds in the ABA group also had lower mRNA levels of the *LlABI3* gene than control seeds (p=0.009).

#### 2.3.3. Relative Expression Level of *LlFUS3*

The analysis of differences in the levels of transcriptional activity of the *LlFUS3* gene (Figure 4) showed a statistically significant interaction effect between all the factors, $F\left(8.24\right)=3.23;p=0.012;\;\eta^{2}=0.52$. The analysis of differences between the tested seed variants showed that the application of phytohormones did not cause significant differences in the transcriptional activity of the *LlFUS3* gene on 15 DAAs and 30 DAAs. Statistically significant effects were observed on 20 DAA. At 4 h, a higher expression level of the *LlFUS3* gene was found in the seeds of plants treated with ABA compared to the control group (p=0.004). In the remaining cases, no significant differences were observed between the compared variants.

#### 2.3.4. Relative Expression Level of *LlBETA*

The analysis of differences in *LlBETA* gene-expression levels (Figure 5) showed a statistically significant interaction effect between all the factors $F\left(8.24\right)=4.73;p=0.001;\;\eta^{2}=0.61$. The level of transcriptional activity of the *LlBETA* gene showed a statistically significant increase with a linear trend during the seed development, which was independent of the phytohormone administered and the time of harvest (Figure 5). All the expression levels of the studied gene on successive days of seed development were statistically significantly higher, at least at the level of p<0.01. On successive days, the amount of *LlBETA* gene transcripts was increasing for each hourly harvest and the tested variants (GA, ABA, Control) (Figure 5). In variant 15 DAA, at the 4th hour of harvest, it was found that the seeds of the plants after GA_3_ application had a significantly higher level of *LlBETA* gene transcripts compared to control seeds (p=0.002) and those treated with ABA (p=0.001). However, in the next stage (20 DAA), the seeds of plants treated with GA_3_ had a significantly lower level of expression of the *LlBETA* gene at 0 h, compared to the seeds of plants treated with ABA (p=0.001) and control (p=0.007). At the 4th hour of harvest on 20 DAA, it was confirmed that the control seeds had a significantly higher mRNA level of the studied gene than the seeds of plants after the application of GA_3_ (p=0.007) and ABA (p=0.008).

On the other hand, on 30 DAA, a statistically significant, lower level of the transcriptional activity of the tested gene was observed in the seeds from the control variant in comparison to the seeds of plants after the application of ABA (p=0.003) and GA_3_ (p=0.002) (Figure 5). At the 4th hour of harvest, an increase in the expression of the examined gene was observed only in the group of seeds of plants treated with GA_3_ compared to the control seeds (p=0.037).

#### 2.3.5. Relative Expression Level of *LlDELTA2*

The analysis of differences in the relative level of transcriptional activity of the *LlDELTA2* gene (Figure 6) showed a statistically significant interaction effect between all the factors, $F\left(8.24\right)=3.18;\;p=0.013;\;\eta^{2}=0.51$. The *LlDELTA2* gene also showed a statistically significant increase in the expression during the seed development, with a linear trend, regardless of the phytohormone administered and the time of harvest. The statistical significance between the examined days was at least p<0.01.

On successive days of the seed development, the expression level of the *LlDELTA2* gene (Figure 6) was statistically significantly higher. The analysis of the differences between seed variants on 15 DAA showed that seeds of GA-treated plants had significantly higher levels of the *LlDELTA2* gene at the fourth hour of harvesting compared to seeds from the control group (p=0.003) and seeds of plants exogenously treated with ABA (p=0.001). On 20 DAA, no significant differences between the samples harvested at 0- and 8-h mark were observed, while significant differences were confirmed at the 4th hour. Control seeds at the 4th hour had significantly higher levels of *LlDELTA2* gene expression compared to the seeds of plants treated with ABA (p=0.001) and GA_3_ (p=0.002). On 30 DAA, significant differences were confirmed only at the 0-h mark, where plant seeds after ABA application had lower levels of *LlDELTA2* transcripts compared to control (p=0.007) and GA-treated seeds (p=0.023).

### 2.4. Accumulation of Yellow Lupine Seed Storage Proteins (SSP)

The level of accumulation of identified proteins in yellow lupin Taper seeds at successive stages of their development (15, 20, 30 DAA) was determined by means of nanoLC-MS/MS analyses. Due to missing data, in order to verify the differences between the protein levels of the variants, a series of Kruskal–Wallis H-tests and Friedman’s ANOVA were performed, which resulted in reduced sample numbers. A threshold of (α=0.05) was used as binding for the interpretation of statistically significant differences between measurements.

#### 2.4.1. Accumulation of β-Conglutins after ABA or GA Application

The analysis of comparisons between hours within each variant in seeds on 15 and 20 DAA showed no statistically significant differences (Figure 7). However, on 30 DAA, it was confirmed that β-conglutins levels were higher at hour 8 compared to 0, and this effect occurred in both control and GA- and ABA-treated seeds (p<0.05). The elevated accumulation of this protein was also noted at 4 h in the seeds of the control plants and those after GA_3_ application, but in comparison with the data collected at 0 h in the oldest seeds, these results did not show statistical significance. The comparison between seeds indicates higher levels of such storage proteins in control seeds than in seeds of ABA-treated plants (p=0.039). No other statistically significant differences were observed at various hours, developmental stages, or the level of applied phytohormone.

#### 2.4.2. Accumulation of δ-Conglutins after ABA or GA_3_ Application

Kruskal–Wallis H tests and an analysis by means of Friedman’s ANOVA analysis of variance to determine the significance of the results of δ-conglutins accumulation in yellow lupin seeds (Figure 8) showed no statistically significant differences between the tested variants, regardless whether the effect of hour or phytohormone application was tested. After the observations of the average amounts of δ-conglutins, it was noticed that its accumulation on 30 DAA was higher in samples from 0- and 8- mark hours, regardless of phytohormone application, whereas at the fourth hour, there was no significant change in the level of δ-conglutins in the seeds of GA-treated plants.

## 3. Discussion

The filling of seeds with storage materials is an incredibly specific and dynamic process involving many genes, proteins, phytohormones and metabolites [22]. Describing the correlation between these factors and their resulting outcomes, such as the level of accumulated proteins or yield efficiency, is a highly intriguing subject. On the one hand, this interest is purely cognitive and is included in the scope of basic research. On the other hand, it has very specific and practical implications. Nutritional value plays a vital role in the development of an embryo; hence this trait has evolved and become established through evolution. However, when human needs are considered, it carries economic and nutritional significance. In order to achieve stable accumulation of storage proteins and optimize the yield, it is essential to have a thorough understanding of the underlying mechanisms that govern these processes. This is particularly crucial in economically important plants, such as yellow lupin.

### 3.1. The Role of GA and ABA in Controlling Seed-Filling Processes

As shown in many studies, there is a positive correlation between FUS3 activity and ABA levels and a negative correlation between FUS3 activity and GA levels [44]. These hormones act in opposite ways during seed maturation, which might be due to the synergistic functionality of GA and PKL in inhibiting the activity of regulatory genes and the positive correlation between these factors and the presence of ABA [53]. It has been shown that only in the presence of ABA is it possible to activate *LAFL* genes. This occurs as a result of the inhibition of PP2C phosphatase activity and unblocking the transcription factors such as ABI3 through their phosphorylation [37]. The results presented in this paper showing the expression of *LlLEC2*, *LlABI3*, *LlFUS3*, *LlBETA* and *LlDELTA2* complete and extend the body of knowledge regarding the function of *LAFL* regulatory genes and storage protein accumulation in yellow lupin.

### 3.2. Transcriptional Activity of the LlLAFL Gene Network in Yellow Lupin

RNA-seq experiments showed that the expression of *LlLEC2* varies among the different research variants (Figure 1a). In the youngest seeds, 10 days after flower formation, the amount of *LlLEC2* transcript was almost undetectable, reaching its highest level on day 20 and decreasing again around 80-fold in seeds on 30 DAA. A slightly different pattern was observed in subsequent experiments using the quantitative PCR technique (Figure 2), in which the amounts of *LlLEC2* systematically decreased to almost undetectable levels in seeds harvested on day 30. These differences may be due to changes in the environmental conditions of field crops in 2018–2020.

While the transcriptome analyses described in this paper were conducted, it was puzzling to notice that there were no homologs of the *LEC1* gene in the cDNA libraries obtained from yellow lupin seeds. This result was also confirmed in independent RNA-seq analyses described by Glazinska [54] and data deposited in the LuluDB database (http://luluseqdb.umk.pl/basic/web/ on 30 October 2020). It turns out that a mutation of the *DCL1* gene, which encodes an enzyme involved in miRNA maturation, results in the inhibition of *LEC1* expression in *A. thaliana*, and in return there is an increase in the number of transcripts of *LEC2* or *FUS3* [55,56]. In the results presented here, a very similar expression pattern to *LlLEC2* was also observed for the *FUS3* homolog (Figure 1c). Other factors that may modulate the activity level of *LlLEC2* or *LlFUS3* genes may be hormones, including ABA and GA [44,57,58]. The control of GA synthesis and degradation of gene expression by FUS3 affects hormone homeostasis, which may translate into the expression pattern of both *LEC2* and *FUS3*. It should be remembered that the mentioned genes and hormones act in specific positive and negative feedback loops. Therefore, it can be speculated that the observed decrease in the expression of *LlLEC2* (Figure 1a) and *LlFUS3* (Figure 1c) may be due to reasons other than direct relationships between them. Perhaps the reasons for the relatively short-lived increase in the expression of *LEC2* and *FUS3* homologs could be the lack of *LEC1* activity in yellow lupin or changes in hormone balance.

*LlABI3*, which belongs to the B3 family of transcription factors, is another element in the network of dependence and control of the accumulation of storage materials in yellow lupin seeds. Detailed studies conducted on the mechanisms of seed maturation in other plant species show that it has a very specific function [59]. ABI3 is expressed at later stages of seed development. Its effects are significantly different from those observed as a result of the activity of *LEC1*, *LEC2* or *FUS3*, which are necessary at earlier stages of seed development [11,38,60]. Of the above-mentioned genes, only *ABI3* is expressed throughout the embryo. The protein it encodes is the only one that contains all four regulatory and DNA-binding motifs characteristic of the B3 type family of transcription factors [26,59,61]. There are increasing reports of the occurrence of different transcriptional variants of this gene in plants such as pea and tomato [62,63]. Studies conducted on lucerne have shown that each of the three transcriptional forms is structurally different, so the encoded proteins may exhibit different modes of action and control the expression of different genes [64]. It has also been confirmed that the transcription factor ABI3 is the most involved in the control of storage-protein accumulation. A study showed that in *Arabidopsis*, FUS3 activity is more important for the lipid-accumulation process [65]. However, it does not mean that ABI3 is an independent activator of genes encoding storage proteins because it is positively controlled by both LEC2 and FUS3.

The analysis of identified promoter sequences of genes encoding storage proteins in plants from different systematic groups shows the presence of additional *cis*-type elements, such as those specific to the *Fabaceae* (E2Fb motif) [66]. This demonstrates the complexity of the overall mechanism of regulation of the processes described. In the study presented in this paper, the expression of *LlABI3* observed in yellow lupin seeds showed the same trend in both the results obtained in the RNA-seq experiment (Figure 1b) and quantitative PCR reactions (Figure 3). The obtained results showed a gradual and systematic increase in the transcriptional activity of the studied gene at successive stages of seed development. Comparing the obtained results of the expression of this gene to the transcriptional activity of the genes belonging to *LlLAFL*, described above in yellow lupin of the Taper variety, a clear shift in time can be observed (Figure 1b). This pattern confirms the involvement of *LlABI3* in processes related to seed filling at the final stages of seed development after the activation by earlier elements such as *LlLEC2* and *LlFUS3*. This increased activity is also significant because of other transformations that can be controlled by *LlABI3*. These undoubtedly include the establishment of dehydration tolerance and seed longevity [30,37,67].

### 3.3. Expression of Genes Encoding β- and δ-Conglutins and the Level of Their Accumulation in Yellow Lupin Seeds

The effect of LAFL activity is an increase in the expression of genes encoding storage proteins. However, there are few literature data on changes in the expression of genes encoding conglutins in yellow lupin. Most studies have been done on narrow-leafed lupin and white lupin. In *L. angustifolius*, 16 genes encoding four major classes of conglutins have been identified, while data available from gene banks indicate that there are nine such genes in *L. albus* [6,7]. Sequencing analyses performed on cDNA libraries from yellow lupin seeds presented in this paper (GSE207091) showed the existence of five genes encoding β-conglutins and one encoding δ-conglutins. However, it is not entirely possible to compare these results with studies conducted on other lupin species, which identified storage proteins accumulated in seeds of plants grown under controlled conditions [6]. The expression patterns of the *LlBETA* and *LlDELTA2* genes indicate that their activity is highest in the oldest seeds (Figure 1e,d). In the 30 DAA variant, the amount of mRNA of all identified conglutins was many times higher than in the other variants. RNA-seq experiments showed that *LlBETA* genes are activated earlier, compared to *LlDELTA2*. Their slight increase in expression was observed in seeds harvested at day 20 (Figure 1d), while the activity of genes encoding δ-conglutins was not yet evident (Figure 1e).

Using the RT-qPCR technique, an increase in the transcriptional activity of both identified genes was observed in the same variant of yellow lupin seeds (Figure 5 and Figure 6). The highest amount of mRNA of genes determined by both RNA-seq and RT-qPCR techniques was observed in the oldest seeds (30 DAA). Interestingly, in yellow lupin, δ-conglutins represent the greater part of the storage proteins. Using the quantitative PCR technique showed that this elevated level was already present in 20 DAA seeds (Figure 6). A similar observation was made in the study by Foley et al. 2015 [7], which showed that yellow lupin seeds harvested at 20–26 DAA have higher levels of δ-conglutin-gene expression than β-conglutin. In other lupin species, these relationships are different and, for example, in narrow-leafed lupin or *L. mutabilis*, β-conglutins are predominant [7]. Confirmation of transcriptomic data and those obtained by quantitative PCR technique was provided by experiments using chromatographic separation of peptides identified as conglutins. In these experiments, the highest accumulation of SSP in the oldest seeds was confirmed (Figure 7 and Figure 8). Detailed analysis also confirmed a higher proportion of δ-conglutins in the total pool of storage proteins.

### 3.4. Effect of GA_3_ on the Expression of LlLAFL and Genes Encoding β- and δ-Conglutins

In the present study, yellow lupin cultivar Taper was treated with GA_3_ and ABA at early stages of seed development (15 DAA). The application of phytohormones in any variant did not dramatically change the expression pattern of the studied genes in the control variant. However, several characteristic changes in the levels of transcriptional activity were observed. For the *LlABI3* gene, the most frequently seen effect of GA_3_ application was an increase in transcriptional activity (Figure 3). Literature data suggest that both LEC2 and ABI3 or FUS3 are involved in inhibiting the expression of genes involved in GA biosynthesis [43,44]. The addition of a further pool of GA_3_ may negatively affect the complete transformation of the seed. It can also result in increased expression of genes encoding factors that inhibit the phytohormone biosynthesis process. Such a mechanism has been described for ABI3. However, in the case of the FUS3 protein, its stabilization in the presence of ABA appears to be crucial [44].

Exogenous GA_3_ application did not stimulate the expression of genes encoding conglutins, although the hormone showed a positive effect on *LlABI3* gene expression (Figure 3) and is involved in the positive control of many growth and developmental processes, especially at earlier stages of seed development. In most of the tested variants of yellow lupin cultivar Taper, GA_3_ administration was associated with a decrease in conglutin gene expression. The only positive effect was observed in the oldest seeds. At hour four, significantly higher mRNA amounts of both conglutins were recorded, and at hour zero, higher expression of genes encoding β-conglutins was identified (Figure 5 and Figure 6). The literature data [7,22,68,69] indicate that in the vast majority of cases, gibberellin negatively affects the function of the gene network controlling seed filling, which indirectly leads to the accumulation of lower amounts of storage proteins. The negative effect of exogenously administered GA was also observed in experiments conducted on rapeseed [70]. In these experiments, the dry weight of seeds, the content of sugars, and the amount of proteins in the oldest seeds increased, but the weight of a thousand seeds and the content of fats decreased. The same experiments also showed a negative effect of gibberellins on the expression of *ABI3* or *LEC2*, which confirms the activity of this hormone in the process of seed filling. The results of our study showed no significant effect of GA_3_ on the expression of all identified *LlLAFL* genes. Thus, changes in the expression of conglutin-encoding genes can hardly be linked to this regulatory system and the effect of exogenous GA_3_ application, directly.

It is well known that phytohormones function in feedback mechanisms, often of a negative nature. Additional factors coordinating these relationships include a photoperiod or simultaneously the quality and quantity of light [71]. It takes several days from the administration of the hormone to the described changes, however, and in many experiments, it has been shown that the level of applied hormones can change and might be maintained in cells for a period of even tens of days. This depends, among other things, on the environmental conditions in which the experiments are conducted and the metabolic state of the cells. Studies conducted on various plant species growing under natural conditions are especially relevant in this context [72,73,74]. It can be concluded that the observed effects are due to the activity of the applied hormone at an early stage of seed development. Indeed, it is possible that gibberellin administered at that time disrupts both hormonal homeostasis and the activity of *LAFL* genes and other factors modulating the seed-filling process. This may be due to altered activity of DELLA proteins (inhibitors of GA action), e.g., RGL3 or elements related to the function of, e.g., auxins, such as YUCCA-like [75,76,77]. Although clear changes in the mRNA levels of conglutin genes were observed in the oldest seeds in selected variants, this did not translate directly into an increase in the amount of proteins. This could be caused by a delay between transcription and translation processes, but could also be due to other reasons, e.g., the lack of opportunity to synthesize storage proteins at this stage of seed development, or the initiation of transcript-degradation processes and the onset of desiccation. More detailed studies and experiments performed under controlled conditions are needed to answer the question of whether in fact the observed increase in conglutins is due to the administration of gibberellins.

### 3.5. Effect of ABA on the Expression of Identified Genes

In none of the studies presented in this paper was it possible to observe positive, statistically significant, and fully reliable relationships between the application of exogenous ABA on the expression of *LlLAFL* genes and genes encoding β- and δ-conglutins. Although literature data indicate that ABA promotes the transcriptional activity of *LEC2*, *ABI3*, or *FUS3* genes [37], no such relationships were observed in yellow lupin seeds. Only incidental increases in the amount of *LlFUS3* (Figure 4) and *LlABI3* (Figure 3) mRNAs were noted in seeds harvested at zero and four hours on 20 DAA. However, they did not show any trend, and despite some repeatability, it cannot be said that these changes were fixed, and they might rather have occurred due to random fluctuations in environmental conditions and the plants’ response to these conditions.

The quality and quantity of light affects the activity and biosynthesis of ABA. Far-red light has been shown to promote abscisic acid accumulation. Temperature can also have similar effects, although this is not a direct effect [78,79]. In the case of the studies described in this paper on yellow lupin, the limited effect of exogenously applied ABA might occur because of the stabilizing function that ABA might have on the activity of e.g., LlFUS3 protein. In that case, abscisic acid affects the stability and activity of the transcription factor, but not the expression of the gene encoding this protein. The results of numerous studies also suggest that the presence of a phytohormone on its own is not sufficient for the changes that ABA response elements may be subject to. In such instances, the presence of additional elements in the form of specific protein activity becomes necessary [44,80].

This study shows the most significant elements of the pathway for control and accumulation of storage proteins in yellow lupin treated with exogenous ABA and GA_3_ application. Although the overall mechanism seems to be similar to those described in other plant species, distinct differences such as the lack of *LEC1* transcriptional activity were also noted. For a full understanding of the course of described events, further comparative studies are needed for both the natural growing systems and variants grown under controlled conditions, as well as a detailed description of all the interactions between the regulatory elements of such a complex and multi-level control network.

## 4. Materials and Methods

### 4.1. Research Material Used in RNA-Seq Experiments

Cultivation of yellow lupin variety Taper was carried out under field conditions in the experimental plots of the Astronomy Center of the Nicolaus Copernicus University in Piwnice near Torun (Poland, 53°05′42.0″ N 18°33′24.6″ E), as described in detail in Glazinska et al. [81]. The research material used for the study consisted of seeds of yellow lupin of the Taper variety harvested on the successive days of their development. RNA-seq experiments included 10, 20 and 30 DAA (day after anthesis), accordingly. Sample pictures showing harvested seeds are shown in Figure 1f. The material was collected in 2017. For each developmental stage, seeds were collected from at least 20 plants from different plots. For a complete understanding of the experiment, its scheme is shown in Appendix A.

### 4.2. Research Material Used in the Experiments of the Impact of ABA and GA_3_ Applications

To study the effect of phytohormone applications (ABA and GA_3_) on the expression of identified genes and the level of accumulation of selected proteins, the following variants were introduced in successive growing seasons (2018–2020): C—control, ABA—plants after abscisic acid application, GA—plants after GA_3_ administration. For this purpose, nine plots were designated to ensure that material could be collected from three independent biological replicates. Each of the separated plots was sprayed with a specific solution of phytohormone ABA or GA_3_ (OlChemIm, Olomouc, Czech Republic) at a concentration of 0.1 mM, with the addition of 0.05% Tween^®^20 (SERVA Electrophoresis GmbH, Heidelberg, Germany). The control plant plots were applied with 0.05% Tween^®^20 solution. Approximately 2.5 L of the prepared solution was used per spray. Phytohormone application was carried out on the 15th day after anthesis (15 DAA). Seeds were harvested just before spraying (0 h, around 8 a.m.) to serve as control samples for phytohormone-treated samples. Subsequent harvests were conducted four (4 h, about 12 p.m.) and eight hours (8 h, about 4 p.m.) after phytohormone application. Seeds were also harvested on the following days of their development: 20 and 30 days after anthesis, at the same hours. Immediately after harvesting, seeds were frozen in liquid nitrogen and stored at −80 °C until the isolation of total RNA. For a complete understanding of the experiment, its scheme is shown in Appendix A.

### 4.3. Identification and Expression of Selected Genes in Yellow Lupin Seeds

#### 4.3.1. Isolation of Total RNA

Total RNA isolated from seeds of the yellow lupin cv. Taper, harvested on 10, 20 and 30 DAA (in June–July 2017) was used for the preparation of transcriptome libraries. After harvesting, the plant material was frozen in liquid nitrogen and stored at −80 °C until RNA extraction. RNA was isolated from about 100 mg of ground tissue using the Quick-RNA^TM^ MiniPrep kit (Zymo Research Corp., Irvine, CA, USA) according to the manufacturer’s instructions. An analogous procedure was followed in order to isolate total RNA used to perform RT-qPCR reactions. The plant material consisted of seeds harvested on 15, 20 and 30 DAA (in 2018–2020). For these cases, total RNA was isolated using the E.Z.N.A.^®^ Plant RNA Kit (Omega Bio-tek, Norcross, GA, USA) in accordance with the producer’s instruction.

The electrophoretic analysis and the spectrophotometric measurement by the Nanodrop ND-1000 spectrophotometer (Thermo Scientific, Waltham, MA, USA) showed that ribonucleic acid was of very good quality and had not degraded. These results were confirmed by automated quantitative and qualitative analysis based on capillary separation, fluorescence quantification, and RNA Integrity Number (RIN) measurements on a BioAnalyzer 2100 (Agilent, Santa Clara, CA, USA) using the Small RNA Kit (Agilent, Santa Clara, CA, USA). All samples had a RIN of around 9 and above and were submitted for library construction and sequencing to OpenExome s.c. (Warsaw, Poland). All steps of the isolation and validation of RNA extracted from yellow lupine seeds for both RNA-seq and RT-qPCR were performed in the same way.

#### 4.3.2. NGS Sequencing of Transcriptomes

A cDNA library was constructed on a validated RNA template using the TruSeq Stranded Total RNA Ribo–Zero Plant depletion kit (Illumina, San Diego, CA, USA). The use of appropriate nucleotide tags at the 5′ and 3′ ends of the obtained fragments allowed for the multiplication and sequencing of the library created in this way. Then, the sequences of the previously obtained cDNA fragments were read in a series of sequencing reactions performed on the HiSeq 4000 platform (Illumina, San Diego, CA, USA). For each research variant (Seeds 10 DAA, 20 DAA and 30 DAA), sequencing was performed for three biological repeats.

197,917,268 reads were used (about 9% of the percentage of reads from each sample) to create a reference transcriptome after low-quality fragments had been filtered out (Trimmomatic tool [82]). The assembly of the prepared reads into a reference genome was carried out using the Trinity version 2.5.1 [83]. In all, 319 392 transcripts were obtained, which were annotated using the BLAST tool [84] with proteins from the SwissProt database and transcripts from the BLAST database—Nucleotide collection—containing GenBank + EMBL + DDBJ + PDB + RefSeq sequences. Each annotation was performed twice: once on proteins/transcripts of all organisms, and the second time on proteins/transcripts filtered for *Lupinus angustifolius*. Only the best matches were reported, taking into account that the e-value should be no less than $1\times 10^{-20}$. The collected reads were linearized to the created reference genome using the Bowtie2 tool [85]. For each biological repeat, the analysis was performed in the same way.

#### 4.3.3. cDNA Identification of Selected Genes

cDNA identification of the *LlLEC2*, *LlABI3*, *LlFUS3*, *LlBETA* and *LlDELTA2* genes was carried out by assembling and comparing reads from the transcriptome identified as fragments of the aforementioned genes found in related species, mainly *Lupinus angustifolius*, the almost complete genome sequence of which is known [86,87,88]. The identified cDNAs have been deposited in the database GEO (GSE207091).

#### 4.3.4. Determination of Expression Levels of Selected Genes

The expression level of the genes studied was estimated for both each individual gene and isoform and described using RPKM (reads per kilobase per million mapped reads). Direct comparisons among the studied variants made it possible to determine the level of statistical significance of the observed differences. Using the Samtools idxstats tool version 1.3.1 [89], expression levels were determined in selected transcripts from seeds of the yellow lupin cv. Taper at 10, 20 and 30 days after flower development. Differences in expression levels of transcripts were analyzed using the edgeR tool version 3.12.1 (exactTest) [90].

#### 4.3.5. Determination of the Expression of Selected Genes by RT-qPCR

To validate the RNA-Seq results, gene transcript levels were determined by reverse transcription-quantitative PCR (RT-qPCR). The reverse transcription reaction (RT) was performed with 1 or 2 µg of the total RNA and the NG dART RT kit (EURx, Gdańsk, Poland). The obtained cDNA was diluted five times and used for the Real Time PCR reaction carried out with the SensiMix™ II Probe Kit (Bioline GmbH., Luckenwalde, Germany) following the manufacturer’s protocol. The LightCycler480 (Roche, Basel, Switzerland) was used to perform the reaction under the following conditions: 95 °C for 10 min, 40 cycles of 95 °C for 10 s, 57 °C for 1 min, and 37 °C for 40 s. The reaction mixture consisted of 5 μL of 2 × SensiMix™ II Probe, 0.2 μM of each primer, 0.1 μM of a specific UPL probe (Roche, Switzerland) and 5 μL of diluted cDNA. Each tested variant had three biological and four technical repeats. The level of activity of *LlLEC2, LlABI3, LlFUS3, LlBETA* and *LlDELTA2* homologs was compared to the level of a reference gene—*ACTIN* (*LlACT*), previously identified in yellow lupine. Gene-specific primers for the tested genes and molecular UPL were designed in the Universal ProbeLibrary (UPL) Probe Finder version 2.48 (Roche, Basel, Switzerland) and listed in Table 1. To determine reaction efficiencies for the genes tested, calibration curves were performed at four measurement points (1 ng/µL; 0.1 ng/µL; 0.01 ng/µL and 0.001 ng/µL). The templates for the reaction and the determination of qPCR efficiency were determined by flanking primers 200–300 nt distant from the primer of the actual amplicon used in the qPCR reaction. The results were analyzed in LightCycler 480 Software version 1.5.1.62 (Roche, Basel, Switzerland) dedicated to programming reaction conditions, depositing data and analyzing the obtained results.

### 4.4. Protein Profiling in Yellow Lupin Seeds

#### 4.4.1. Protein Isolation

Protein isolation was performed with the E.Z.N.A.^®^ Plant RNA Kit (Omega Bio-Tek, Norcross, GA, USA) following the manufacturer’s protocol. Isolated proteins were suspended in 200 mM HEPES buffer (Thermo Scientific, USA) (pH 8.5). The amount of obtained proteins was checked by spectrophotometric measurement under UV light at 280 nm using a Nanodrop ND-1000 spectrophotometer (Thermo Scientific, USA).

#### 4.4.2. Reduction, Alkylation, Digestion, and TMT Labeling

The FASP (Filter-Aided Sample Preparation) method was used to prepare samples from *Lupinus luteus* seeds for mass spectrometric analysis [91]. In all, 80µg of protein isolates samples were subjected to reduction, alkylation, and digestion on centrifugal filters with a 10 kDa molecular weight cut-off. The filter was conditioned by rinsing with 500 µL of MiliQ water and 100 µL of 100 mM TEAB (triethyl ammonium bicarbonate, Thermo Scientific, USA) buffer by spinning the filter after each application of the solution (30 min, 10,000× *g*, 4 °C). Samples were reduced with 10 mM TCEP (tris(2-carboxyethyl) phosphine, Thermo Scientific, USA) for 1 h in 55 °C and subsequently alkylated with 17 mM iodoacetamide (Thermo Scientific, USA) in 100 mM TEAB for 30 min. at room temperature. After incubation, the samples were centrifuged (30 min, 10,000× *g*, 4°C) and digested with trypsin solution (protein: enzyme, 40:1) in 50 mM TEAB (Promega Corp., Madison, WI, USA) at 37 °C for 18 h. Peptides were labeled with the TMTsixplex Label Reagent Set (Thermo Scientific, USA) according to the manufacturer’s instructions. Due to the number of the samples (21 samples in 3 biological replicates each), a total of 13 independent labelling procedures were performed. Mixed isolates from 9 selected samples equilibrated and diluted to a concentration of 320 µg/mL were used as a reference sample (added to each measurement). In every labelling, TMT channel 126 was occupied by one reference sample, which allowed the comparison between LC-MS runs. The resulting samples were stored at −80 °C until further processing. All reagents and solvents were suitable for LC-MS analysis. 

#### 4.4.3. Chromatographic Separation of Proteins by nanoLC-MS/MS Method

The nanoLC-MS/MS measurements were carried out on the LTQ Elite Orbitrap ETD (Thermo Scientific, USA) connected to the Easy nLC 1000 chromatograph (Thermo Scientific, USA). Peptides were trapped using Acclaim PepMap C18 2 cm trap column (Thermo Scientific) and separated on Acclaim PepMap C18, 100 A, 500 mm × 0.075 mm × 3 μm (Thermo Scientific, USA) in ambient temperature. Mobile phases A and B were 0.1% formic acid in water and 0.1% formic acid in acetonitrile: water 90:10 (*v*/*v*). Chromatographic flow was set to 300 nL/min with gradient 2–55% phase B in 150 min. Volume of injection was set to 5 µL. The external calibration of mass spectrometer with LTQ Velos Positive calibration standard was employed, with resulting SD < 1 ppm. Measurements were performed in a positive ion mode in a data-dependent manner in MS range of 110–2000 *m*/*z*. Capillary voltage was set to 3 kV. HCD fragmentation of top 10 peaks was employed, with normalized collision energy set to 35 eV in 1 *m*/*z* isolation window with minimum 2+ charge state of parent ion and dynamic exclusion for 30 s after 2 spectra. 

#### 4.4.4. Protein Identification and Quantification 

Mass spectra were processed with Proteome Discoverer 2.4. Sequest HT (Thermo Fisher Scientific), and Mascot search engines (http://www.matrixscience.com) accessed on 30 September 2019 were used to extract and annotate MS/MS spectra. Mascot searches were conducted by means of SwissProt database (30 September 2019) with Viridiplantae taxonomy with enzyme trypsin (2 missed cleavages), maximum precursor error of 20 ppm, maximum fragment error of 0.1 Da and TMT 6 plex and carbamidomethyl static modifications as well as N-term acetyl and methionine oxidation as dynamic modifications. Sequest HT search was conducted on custom database (SwissProt and TrEMBL sequences for *Lupinus* taxonomy including subtaxonomies) downloaded from UniprotKB on 30 September 2019. Trypsin was selected as an enzyme, with a maximum of 2 missed cleavages. Maximum precursor and fragment errors were set to 20 ppm and 0.1 Da respectively. Modifications were set in the same manner as for the Mascot search. Reporter ion quantification was performed using Proteome Discoverer 2.4 software (Thermo Fisher Scientific). Spectra were normalized to the total amount of peptides in the sample and scaled to the reference sample channel (TMT 126). Unique and razor peptides were used for quantification, with at least two peptide matches per protein. The false discovery rate was set to 0.01 (strict) and 0.05 (relaxed). Shared peptides were not excluded from the quantification results, as they were later used to estimate the total amount of specific protein groups, e.g., conglutins. The mass spectrometry proteomics data have been deposited to the ProteomeXchange Consortium via the PRIDE (1) partner repository with dataset identifier PXD044299 accessed on 2 August 2023.

## 5. Conclusions

The performed expression studies of the identified genes indicate that a network of the *LlLAFL* family (*LlLEC2*, *LlABI3*, *LlFUS3*) has an impact on the process of maturation of seeds of the yellow lupin of Taper variety. Efficient accumulation of conglutins in the seeds is the result of this network’s influence. Thus, it can be concluded that the overall expression pattern of genes encoding storage proteins and *LlLAFL* genes obtained for the seeds of yellow lupin of the Taper variety is consistent with literature data describing the operation of these mechanisms in other plant species, including crop plants. Although the basic mechanism is common, i.e., the increase in conglutin gene expression at subsequent stages of seed development and the succession of *LAFL* network gene expression, there are some differences in the pattern specific to yellow lupin. These differences include the lack of expression of the *LEC1* homolog and the amount of accumulated storage proteins. In the present study, it was shown that the predominant form of seed storage proteins is δ-conglutin, which differs from observations made for other species in which β-conglutin was predominant. All these relationships have not yet been described for yellow lupin cv. Taper. A better understanding of how these mechanisms work requires further research.

The conducted experiments show that the exogenous application of ABA and GA_3_ is not a key factor determining changes in the expression of *LlLAFL* genes in the conditions of natural cultivation of yellow lupine. Exogenously administered GA_3_ in the initial stages of seed filling had a positive effect only on the expression of genes encoding conglutins on the 30th day after anthesis, showing no changes in the accumulation of storage proteins.

This research expands current knowledge of the participation of ABA and GA during yellow lupin seed development and provides clues for further attempts to use phytohormones in the regulation of processes related to the accumulation of storage materials. So far, there has been no research on the effects of these phytohormones on the regulation of seed filling in lupins under natural conditions.

## Figures and Tables

**Figure 1 ijms-24-12380-f001:**
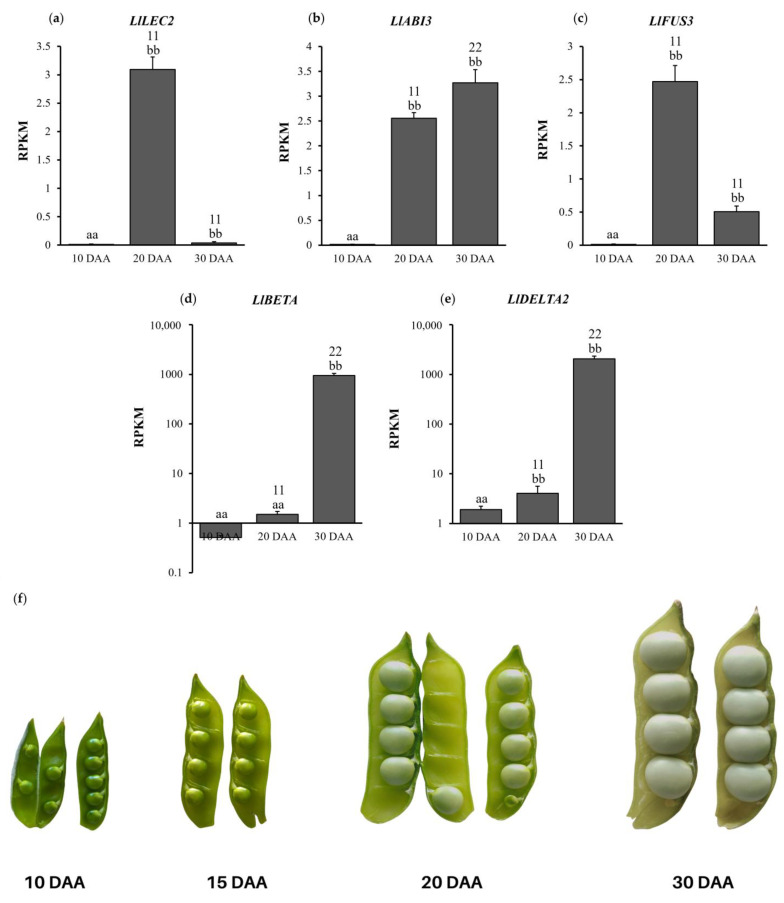
Comparison of changes in expression level of selected genes in successive days of yellow lupine seed development obtained by RNA-seq: Expression level of (**a**) *LlLEC2*; (**b**) *LlABI3*; (**c**) *LlFUS3*; (**d**) *LlBETA*; (**e**) *LlDELTA2*; (**f**) Pods and seeds of yellow lupin cv. Taper grown under natural conditions, showing the studied stages of seed development: 10, 15, 20, 30 DAA; DAA—Day after anthesis; statistical significance determinations: aa—*p* ≥ 0.01—no statistical significance for the compared experimental variants; bb—*p* ≤ 0.01—observed differences show statistical significance of the indicated samples (20 DAA, 30 DAA) relative to the 10 DAA variant; 11/22—*p* ≤ the observed difference between the 20 DAA vs. 30 DAA variants shows statistical significance.

**Figure 2 ijms-24-12380-f002:**
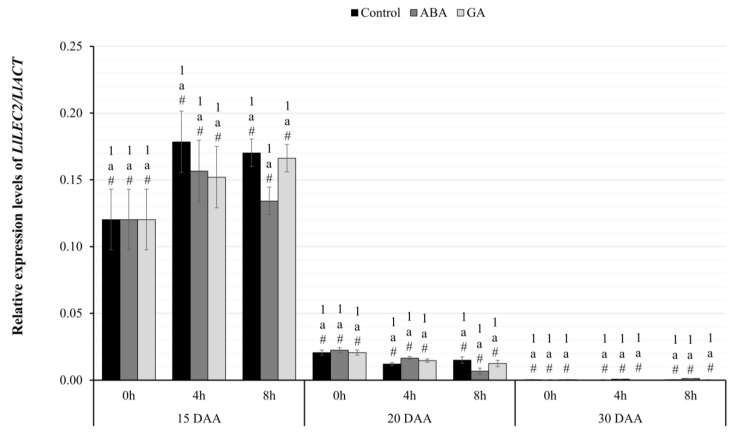
Relative transcriptional activity of the *LlLEC2* gene in the seeds of yellow lupine cv. Taper in relation to the reference gene *LlACT* in the successive days of development after the application of ABA or GA_3_ at 0, 4, and 8 h. Abbreviations: DAA—Day After Anthesis, ABA—plant seeds after abscisic acid application, GA—plant seeds after gibberellin (GA_3_) application. Measurement errors represent 95% confidence intervals for the collected data. Statistical significance determinations: a—simple effects for differences between developmental stages for individual hours of harvesting; 1—simple effects for differences between hours of harvesting within a given stage of development; #—simple effects for differences between seed variants (C/ABA/GA) within a particular harvest hour and development stage.

**Figure 3 ijms-24-12380-f003:**
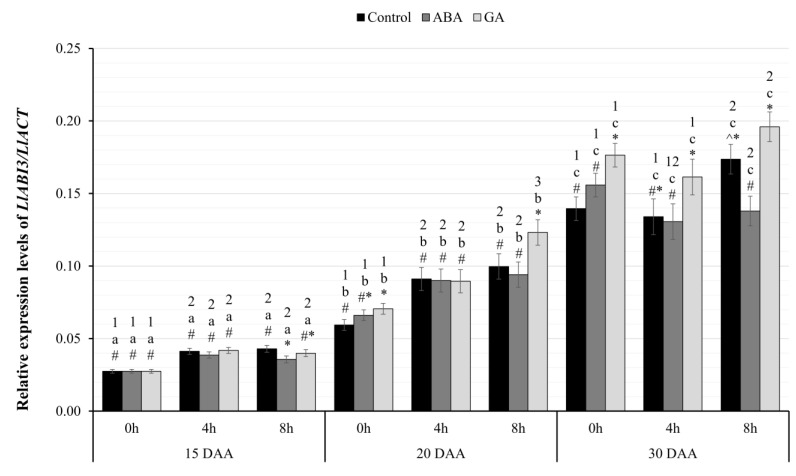
Relative transcriptional activity of the *LlABI3* gene in the seeds of yellow lupine cv. Taper in relation to the reference gene *LlACT* in the successive days of development after the application of ABA or GA_3_ at 0, 4, and 8 h. Abbreviations: DAA—day after anthesis, ABA—plant seeds after abscisic acid application, GA—plant seeds after gibberellin (GA_3_) application. Measurement errors represent 95% confidence intervals for the collected data. Statistical significance determinations: a, b, c—simple effects for differences between developmental stages for individual hours of harvesting; 1, 2, 3—simple effects for differences between hours of harvesting within a given stage of development; *, ^, #—simple effects for differences between seed variants (C/ABA/GA) within a particular harvest hour and development stage.

**Figure 4 ijms-24-12380-f004:**
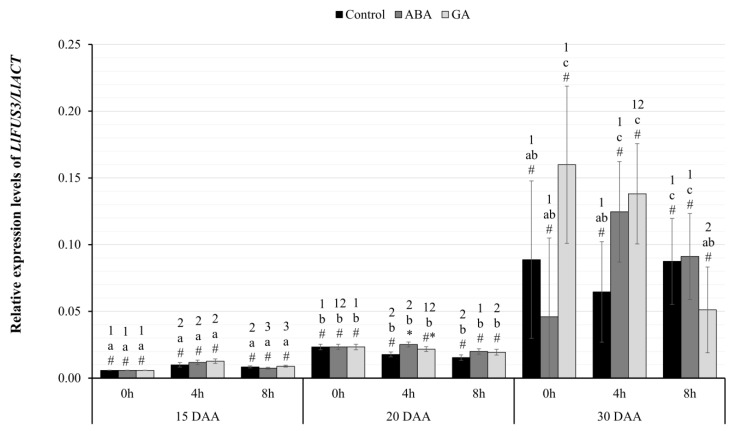
Relative transcriptional activity of the *LlFUS3* gene in the seeds of yellow lupine cv. Taper in relation to the reference gene *LlACT* in the successive days of development after the application of ABA or GA_3_ at 0, 4, and 8 h. Abbreviations: DAA—Day After Anthesis, ABA—plant seeds after abscisic acid application, GA—plant seeds after gibberellin (GA_3_) application. Measurement errors represent 95% confidence intervals for the collected data. Statistical significance determinations: a, b, c—simple effects for differences between developmental stages for individual hours of harvesting; 1, 2, 3—simple effects for differences between hours of harvesting within a given stage of development; *, #—simple effects for differences between seed variants (C/ABA/GA) within a particular harvest hour and development stage.

**Figure 5 ijms-24-12380-f005:**
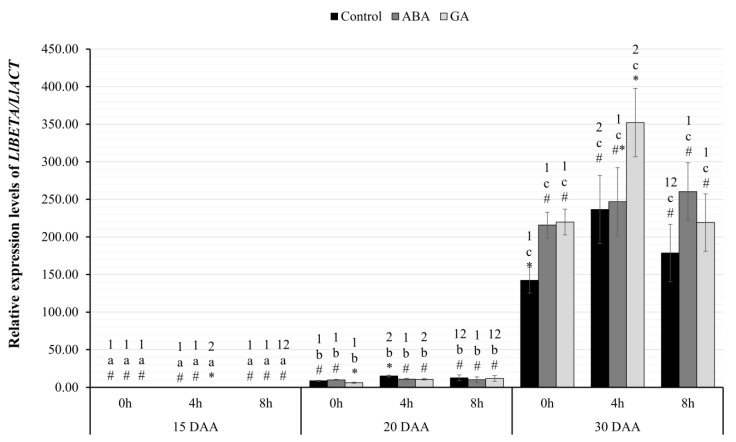
Relative transcriptional activity of the *LlBETA* gene in the seeds of yellow lupine cv. Taper in relation to the reference gene *LlACT* in the successive days of development after the application of ABA or GA_3_ at 0, 4, and 8 h. Abbreviations: DAA—day after anthesis, ABA—plant seeds after abscisic acid application, GA—plant seeds after gibberellin (GA_3_) application. Measurement errors represent 95% confidence intervals for the collected data. Statistical significance determinations: a, b, c—simple effects for differences between developmental stages for individual hours of harvesting; 1, 2—simple effects for differences between hours of harvesting within a given stage of development; *, #—simple effects for differences between seed variants (C/ABA/GA) within a particular harvest hour and development stage.

**Figure 6 ijms-24-12380-f006:**
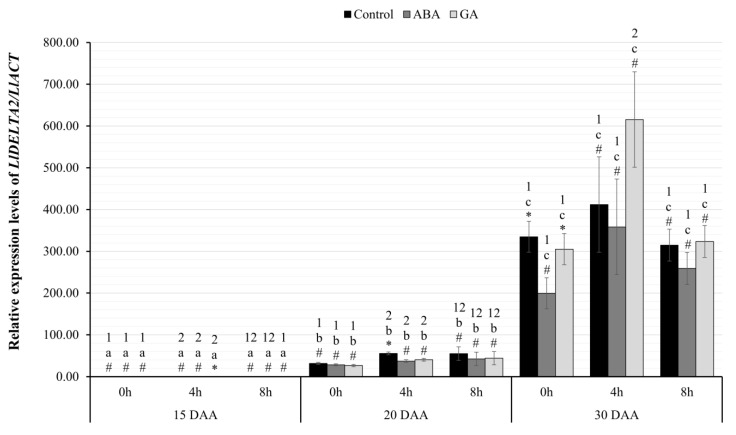
Relative transcriptional activity of the *LlDELTA2* gene in the seeds of yellow lupine cv. Taper in relation to the reference gene *LlACT* in the successive days of development after the application of ABA or GA_3_ at 0, 4, and 8 h. Abbreviations: DAA—day after snthesis, ABA—plant seeds after abscisic acid application, GA—plant seeds after gibberellin (GA_3_) application. Measurement errors represent 95% confidence intervals for the collected data. Statistical significance determinations: a, b, c—simple effects for differences between developmental stages for individual hours of harvesting; 1, 2—simple effects for differences between hours of harvesting within a given stage of development; *, #—simple effects for differences between seed variants (C/ABA/GA) within a particular harvest hour and development stage.

**Figure 7 ijms-24-12380-f007:**
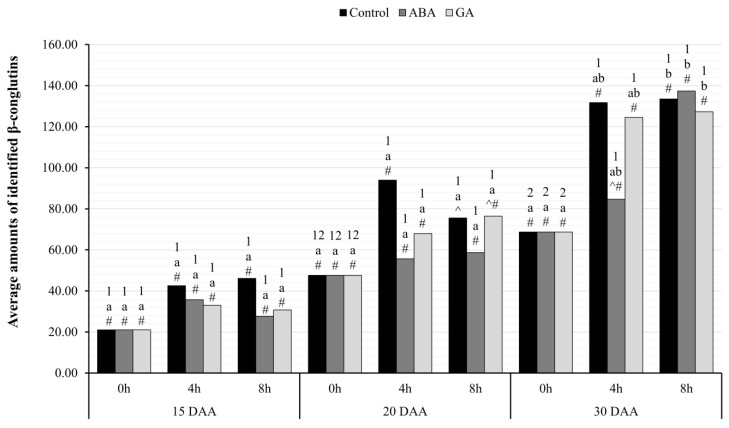
Accumulation level of β-conglutins identified in the seeds of yellow lupine variety Taper on successive days of their development after application of ABA or GA, collected at hours: 0, 4, and 8. Statistical significance determinations for paired comparisons: a, b—simple effects for differences between hours for individual harvest hours; 1, 2—simple effects for differences between development days within the development stage; ^, #—simple effects for differences between seed variants (C/ABA/GA) within a particular harvest hour and development stage.

**Figure 8 ijms-24-12380-f008:**
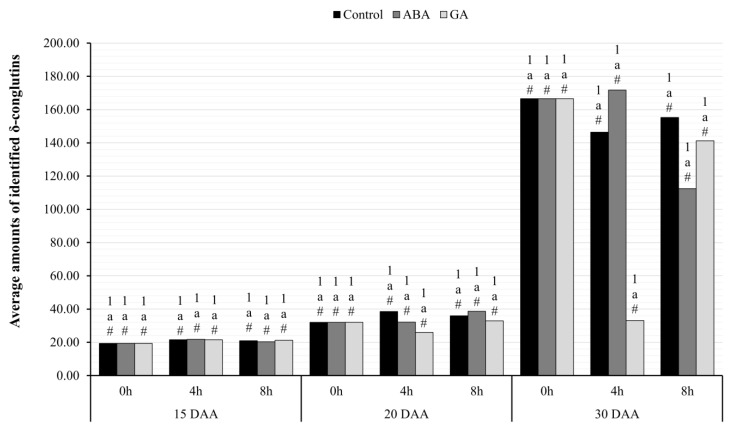
Accumulation level of δ-conglutins identified in the seeds of yellow lupine variety Taper on successive days of their development after application of ABA or GA, collected at hours: 0, 4, and 8. Statistical significance determinations for paired comparisons: a—simple effects for differences between hours for individual harvest hours; 1—simple effects for differences between development days within the development stage; #—simple effects for differences between seed variants (C/ABA/GA) within a particular harvest hour and development stage.

**Table 1 ijms-24-12380-t001:** List of specific primers and UPL probes used in RT-qPCR reaction.

Gene Tested	Sequence 5’→3’		Length (nt)	Melting Temp.(°C)	UPL Probe Number
*LlLEC2*	FP	GCCGGATTATTATCCCAAAGA	21	50.5	30
	RP	TTCCTTTTTGCAAAGGGTTG	20	47.7	
*LlABI3*	FP	CTATGGCACAGGTGGTTCCT	20	53.8	138
	RP	CTGGGTTTGCATGGCAGT	18	50.3	
*LlFUS3*	FP	CCACCACCTCCACCATTT	18	50.3	69
	RP	TTTCACGAGCCGGAGATG	18	50.3	
*LlBETA*	FP	TGGATTTGGCATAAATGCTG	20	47.7	6
	RP	ACATTGTCTTCAGAACCTGCAA	22	51.1	
*LlDELTA2*	FP	AAGATGATTCAGCAGGAGCAA	21	50.5	69
	RP	TTCCTGCTATGTCCAACAACA	21	50.5	

## Data Availability

RNA-seqdata are deposited in https://www.ncbi.nlm.nih.gov/geo/ (GSE207091—Transcriptional activity of genes in yellow lupine seeds cultivar Taper) accessed on 2 July 2022; The mass spectrometry proteomics data are deposited to the ProteomeXchange Consortium via the PRIDE partner (1) repository with dataset identifier PXD044299 (Effect of ABA and GA application on the accumulation of storage proteins in developing seeds of yellow lupine cultivar Taper) accessed on 28 July 2023.

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
