# Peer review of "Regulatory Effects of ABA and GA on the Expression of Conglutin Genes and LAFL Network Genes in Yellow Lupine (Lupinus luteus L.) Seeds"

_ijms, 2023, doi:10.3390/ijms241512380_

Round 1

Reviewer 1 Report

title of MS can be revised as "Regulatory Effects of ABA and GA on the Expression of en-2 Conglutin Genes and LAFL Network Genes in Yellow Lupine 3 (Lupinus luteus L.) Seeds"

conclusion part is missing hence author should provide the conclusion of the study as per the journal requirement

Author Response

Dear Reviewer,

Thank you very much for taking the time to write a review. The comments received are valuable and will certainly enhance the quality of this manuscript and allow a better understanding of the purpose of the undertaken research. Below I attach our responses to the comments received.

Point 1: Title of MS can be revised as "Regulatory Effects of ABA and GA on the Expression of en-2 Conglutin Genes and LAFL Network Genes in Yellow Lupine 3 (Lupinus luteus L.) Seeds"

Response 1: Indeed, the proposed title will be more readable and relevant to the content of the article (lines 2 - 4).

Point 2: Conclusion part is missing hence author should provide the conclusion of the study as per the journal requirement

Response 2: The manuscript has been expanded with a summary including conclusions (p. 5, lines 794 – 821).

The evaluation of the manuscript notes that the description of the methods could be improved.

The methods used in our study are now standard in modern molecular biology. If there is an ambiguity in any of the methods described, please indicate the exact parts. In addition, the revised manuscript has been supplemented with diagrams (Figure S1, Figure S2) and photographs showing the plant material used for the isolation of RNA and proteins (Figure 1f). We hope these changes will allow you to see the procedure of the used research methodology more clearly and provide details on the data recorded in the introduction of the manuscript.

The evaluation also pointed out the need to improve the support of the results with conclusions.
A summary with conclusions has been added to the revised manuscript.

Reviewer 2 Report

Referee’s comments on the manuscript entitled "The influence of ABA and GA on the expression of genes encoding conglutins and LAFL network genes in yellow lupine (Lupinus luteus L.) seeds" submitted for publication in the IJMS (MDPI), by Natalia Klajn et al. (ijms-2507963).

The objective of the present study is to examine the effect of exogenous hormones treatment on the expression of several genes involved in seed maturation, and the content of accumulated storage proteins in seeds of yellow lupin. However, the manuscript lacks novelty and the data is not strong enough. Authors should enhance the quality of their work, especially in data visualization.

Author Response

Dear Reviewer,

Thank you very much for taking the time to write a review. The comments received are valuable and will certainly enhance the quality of this manuscript and allow a better understanding of the purpose of the undertaken research. Below I attach our responses to the comments received.

Point 1: The objective of the present study is to examine the effect of exogenous hormones treatment on the expression of several genes involved in seed maturation, and the content of accumulated storage proteins in seeds of yellow lupin. However, the manuscript lacks novelty and the data is not strong enough.

Response 1: Previous studies on yellow lupin did not include the analyses of protein accumulation and changes in the transcriptional activity of genes encoding conglutinins and LAFL network genes, and the effect of exogenously applied phytohormones on these processes. Hence, the conducted research complements existing knowledge and presents a new approach in the study of storage protein accumulation in legumes.

Point 2: Authors should enhance the quality of their work, especially in data visualization.

Response 2: The results presented are standard in publications on gene expression or protein accumulation. The graphs have been changed to be more readable and the juxtaposition of more data in a single graph is intended to illustrate the totality of the changes that were observed in our experiment.

In the evaluation of the manuscript, mentioned points for improvement included, among others, the background of the introduction and the references cited. The introduction and the bibliography collected and presented reflect the current state of knowledge on the presented subject matter. If an issue was described too poorly, please indicate the part.

It was also pointed out that the description of methods could be improved. The revised manuscript was supplemented with diagrams (Figure S1, Figure S2) and photos showing the plant material used for the isolation of RNA and proteins (Figure 1f). Hopefully, this will make the procedure in the applied research method easier to understand and will also detail the data recorded in the introduction of the manuscript.

The readability of the results has been improved by making changes to the design of the graphs. However, due to the complexity of the experimental systems, they could not be simplified more in order to present the entirety of the changes observed in the carried out experiments.

The evaluation also pointed out the need to improve the support of the results with conclusions.
A summary with conclusions has been added to the revised manuscript.

We hope that the changes made will allow a better understanding of the profile of our research and the obtained results.

Round 2

Reviewer 2 Report

Comments for second review:

1. What does relative expression (log2FC) mean in Fig1? It should be the RPKM values or the normalized values of RPKM?

2. Table 1. should be a three-line table.

3. Why Fig7 and Fig8 presented as means without standard deviations?

Author Response

Dear Reviewer,

We would like to thank you for providing us with your constructive and detailed review comments on our manuscript. The recommendations and advice have helped us to significantly enhance the quality of the manuscript. We have revised our manuscript according to all the comments. The changes in the Microsoft Word file have been made with the "track changes" feature. Below I attach our responses to the received comments.

Point 1: What does relative expression (log2FC) mean in Fig1? Should it be the RPKM values or the normalized values of RPKM?

Response 1: Thank you for this comment. Indeed, the RPKM values are more appropriate to illustrate the expression levels of the tested genes.

Point 2:  Table 1. should be a three-line table.

Response 2: Please forgive this oversight on our part. The table has been corrected.

Point 3:  Why are Fig7 and Fig8 presented as means without standard deviations?

Response 3: The results presented in Figures 7 and 8 are cumulative averages from nanoLC-MS/MS analyses and illustrate trends in the changes in the amounts of the two types of conglutinins of the tested seed variants. Hence, the use of standard deviation is not justified in this case.

Some minor corrections are also evident in the tracked manuscript. In the "Introduction" section, the rationale for the research undertaken was added and the bibliography was enriched with 22 references (numbers: [3, 4, 9, 10, 14, 15, 16, 19, 20, 21, 27, 28, 29, 31, 32, 34, 35, 36, 42, 46, 47, 50]).

In the "Materials and Methods" and "Results" sections, modifications were made to single sentences related to the use of RPKM values to represent gene expression levels instead of log2FC.

We hope that the changes made are sufficient and will allow our manuscript to be accepted.
